# Cross-Cultural Comparison of Burnout, Insomnia and Turnover Intention Among Nurses in Eastern and Western Cultures During the COVID-19 Pandemic: Protective and Risk Factors

**DOI:** 10.3390/nursrep15020052

**Published:** 2025-02-03

**Authors:** Eveline Frey, Yuen-Yu Chong, Wai-Tong Chien, Andrew T. Gloster

**Affiliations:** 1Faculty of Psychology, University of Basel, 4055 Basel, Switzerland; 2The Nethersole School of Nursing, Faculty of Medicine, The Chinese University of Hong Kong, Hong Kong SAR, China; conniechong@cuhk.edu.hk (Y.-Y.C.); wtchien@cuhk.edu.hk (W.-T.C.); 3Division of Clinical Psychology, University of Lucerne, 6002 Lucerne, Switzerland; andrew.gloster@unilu.ch

**Keywords:** burnout, nurses, cross-cultural comparison, psychological flexibility, COVID, mental health, turnover intention, insomnia, health promotion

## Abstract

**Background/Objectives**: The COVID-19 pandemic has adversely impacted the mental health of nurses worldwide. Nurse burnout results from chronic workplace stress and is characterized by exhaustion, negative or cynical feelings about work, and a diminished sense of accomplishment. This can lead to turnover intention. Therefore, psychological capacities, such as psychological flexibility, that could help nurses regulate and minimize the impact should be studied. This study aimed to compare burnout, insomnia and turnover intention among nurses from an Eastern and Western cultural context and to investigate the role of psychological flexibility as a protective factor against mental health and related problems such as insomnia and turnover intention. **Methods**: Nurses from Hong Kong (n = 158) and Switzerland (n = 294) involved in patient care during the COVID-19 pandemic participated in an online mental health survey. **Results**: We observed high levels of burnout, subthreshold insomnia and turnover intention in nurses from both Switzerland and Hong Kong, with higher burnout rates among Hong Kong nurses and higher turnover intentions among nurses from Switzerland, and that psychological flexibility is a crucial factor that may protect nurses from burnout and insomnia. **Conclusions**: The nurses from both regions involved in patient care one year after the onset of the pandemic suffered from poor mental health. Psychological flexibility was identified as a critical factor in improving nurses’ mental health. The results of this study should be incorporated into health promotions for nursing professionals and help develop specific and effective interventions for practical nursing application.

## 1. Introduction

Healthcare professionals, particularly nurses, face significant mental and physical burdens in their profession, with burnout being one of the most frequent reported consequences [1,2,3]. Burnout, a psychological syndrome resulting from prolonged occupational stress, is characterized by three dimensions: emotional exhaustion, depersonalization and reduced professional efficacy [4,5]. Emotional exhaustion manifests as chronic fatigue and stress, while depersonalization reflects a detached and impersonal approach to others. Reduced professional efficacy denotes a diminished sense of accomplishment in one’s work [3,5,6,7].

High burnout rates among nurses were found even before the COVID-19 pandemic, with a meta-analysis reporting a global prevalence of burnout symptoms at 11.2% [8]. During the pandemic, burnout rates increased greatly, particularly among emergency nurses, with nearly 58% reporting high emotional exhaustion compared to 37% pre-pandemic [9,10]. Sleep disorders, another prevalent issue, affected 38% of healthcare workers during the pandemic, with frontline nurses showing the highest rates [11,12].

The mental and physical health challenges not only impact individual nurses but also have systemic consequences, including increased turnover intentions. Pre-pandemic studies revealed a pooled turnover rate of 18% among nurses [13], which surged post-pandemic, with some studies reporting turnover intentions as high as 78% [14,15,16,17]. These high turnover intentions pose a significant risk to the projected global shortage, with the World Health Organization forecasting a deficit of 7.6 million nurses by 2030 [18].

Therefore, it is important to know both risk and protective factors that contribute to improving mental and physical health of nurses to increase the nurses’ commitment to their job and to counteract the current tendency of resigning or leaving the nursing profession at an early stage [19].

### 1.1. Cross-Cultural Comparison of Mental Health of Nurses

Context specific factors such as cultural, social or work-related aspects influence the mental health of nursing professionals, contributing to regional variations in well-being. Poor working conditions—high workload, long hours, low pay, lack of resources, and insufficient legal regulation—negatively impact nurses’ mental health. The pandemic exacerbated these conditions, with increased workloads, fear of infection, and insufficient support affecting healthcare workers. A cross-country study found that nurses in regions with inadequate protective equipment, poor staff training and limited mental support reported worse mental health than those in better supported regions [1,20,21,22,23].

Furthermore, cultural differences, such as varying values and norms, can also influence the mental health of nurses in different regions. While in individualistic cultures where the self is a central aspect in society, values such as individual rights, concern for oneself and immediate family, autonomy and personal self-fulfillment are significant, values such as duty and obligations to the ingroup, interdependence on other individuals within the group and fulfillment of social roles are significant in collectivistic cultures where the ingroup forms the central unit of society [24,25].

For example, it is conceivable that in collectivistic cultures, mindsets toward the profession may differ from those in individualistic cultures. In collectivist cultures, where the well-being of the community is a central value, this may lead to greater social pressure on nursing professionals to sacrifice themselves for their profession. In Western cultures, which are more individualistic, there is a greater recognition of personal needs and self-care. As a result, nurses in Western cultures may become more aware of their mental health and more willing to set boundaries or seek help to protect their mental well-being.

Additionally, different approaches to the topic of mental health and the associated stigma can impact the mental health of nurses. Nurses from cultures in which mental health is associated with greater stigma may be less open to talk about psychological burdens, as it could be seen as sign of weakness [25]. That could result in less access to mental health promotion programs in these cultures.

Understanding these context-specific factors is crucial for tailoring health programs to regional needs. Despite its importance, cross-cultural studies on nurses’ mental health remain limited, with existing research primarily comparing Western countries. Only a few studies have directly compared burnout rates between Western and Eastern cultures. The results of the studies indicate that differences between cultures exists; however, some studies found higher burnout values for Western countries, while others found higher values for Eastern countries [26,27,28].

This study aims to gain further insight with a larger sample into whether differences in burnout exist across cultures and compare these findings with previous research. Having more knowledge about cross-cultural differences also allows psychological interventions to be tailored to the respective culture, which may be more appropriate than universal interventions.

### 1.2. Risk Factors: Stress, Job Satisfaction, Age and Gender

Encountering stress is consistently linked with increased risk for adverse mental and physical health outcomes, as well as greater utilization of healthcare services [29]. The popular Job Demand Control Support model (JDCM) [30] is a way to explain why nurses often experience stress in their work. It postulates that high job demands, such as being constantly exposed to emotionally draining stressors in the provision of patient care; a hectic workspace with fast-changing and unpredictable situations; and systemic issues such as irregular working hours, “voluntary” overtime, rotating shifts, and understaffing [8,31], low job control and low social support, lead to stress experiences among nursing professionals [32]. The outbreak of the COVID-19 pandemic was an additional stressor for nurses: the workload increased due to high hospitalization numbers; work conditions were restricted for safety; and many experienced feelings of uncertainty, worry, and anxiety about becoming ill due to close contact with infected patients. It can therefore be assumed that the increased job demands have also led to a decrease in the sense of perceived control and support from supervisors and team colleagues. Nurses were partially reassigned from their usual unit to support other units that were treating COVID-19 patients. In addition, nurses did not know how long this situation would last, and due to shortages of nursing staff, there were few resources to support each other or provide additional staff for overloaded departments with a lack of nursing staff.

Besides stress, it is also known that low job satisfaction has a negative impact on mental health. Job satisfaction is the affective orientation that an employee has toward their job [33] and, when compromised, is a well-known risk factor for negative health outcomes, such as burnout [34], and for organizational outcomes, such as absenteeism, intention to quit and turnover [35].

Additionally, age and gender have been identified as risk factors for various mental health problems among healthcare workers, including anxiety, depression, burnout, stress, or sleep disorders. Female and younger healthcare workers are particularly vulnerable to psychological symptoms [36,37,38].

These risk factors are important and well-researched. However, it is difficult to influence them with health-promoting interventions, as stress is part of the nursing profession and job satisfaction is often related to structural and systemic factors that are not directly changeable through health interventions. Therefore, more research is needed on processes that can be more easily changed and implemented with interventions.

### 1.3. Protective Factor: Can Psychological Flexibility Act as a Buffer to Reduce Mental and Physical Harm?

Although some protective factors, such as having social and family support, quality of work environment and career and financial stability, are known [39,40,41], most of these factors largely depend on external conditions and are therefore not easily modifiable by the affected individual. Therefore, there is a need for protective factors that are independent of external conditions and can be promoted and modified through psychological interventions. One potential protective factor that can be modified by psychological interventions is psychological flexibility. Psychological flexibility, a transdiagnostic concept that contains inter- and intra-personal skills, can be defined as the ability to “recognize and adapt to various situational demands; shift mindsets or behavioral repertoires when these strategies compromise personal and social functioning; maintain balance among important life domains; and be aware, open and committed to behaviors that are congruent with deeply held values” [42]. Psychological flexibility has been identified as a psychological factor that can be actively modified by targeted interventions [43,44,45] and has been found to positively affect mental well-being in clinical and non-clinical settings [42,44,45,46]. Moreover, a study conducted during the COVID-19 pandemic identified psychological flexibility as one of the few consistent protective factors against stress, and other studies showed that psychological flexibility could mitigate the impact of ill perceptions toward COVID-19 on mental health and had a significant mediating role over and above the contributions of other known coping-style variables, such as avoidance, positive thinking, seeking social support, and problem-solving [47,48]. With respect to burnout, interventions targeting psychological flexibility have shown promising results in general samples [45] and specifically in a sample of undergraduate nursing students with high avoidance [49].

To our knowledge, however, psychological flexibility has not been examined as a protective factor for burnout, insomnia and turnover intention among nurses during the pandemic. Further, it has not yet been investigated how the effect of psychological flexibility varies in working environments across different health care systems that are inherent in different countries.

This study had two objectives: first, to examine the influence of the context-specific factor of culture on burnout, insomnia and turnover intention rates of nurses. Based on the findings of previous research that showed culture-dependent variations in cognitive, behavioral and emotional components of the personal system [25], we hypothesized that burnout rates, insomnia and turnover intention differ between nurses from Hong Kong with a collectivistic background from Swiss nurses with an individualistic background.

Second, the study aimed to examine the role of psychological flexibility as a protective factor against burnout, insomnia and turnover intention. We hypothesized that psychological flexibility would be a significant protective factor against burnout, insomnia and turnover intention after controlling for stress, job satisfaction, age and gender, which have been identified as risk factors for increased vulnerability for psychological symptoms.

## 2. Materials and Methods

### 2.1. Participants and Procedure

The two-region online survey was completed by 452 participants from Hong Kong (n = 158) and Switzerland (n = 294) [50]. The survey took place during the second wave of the COVID-19 pandemic, between January and March 2021. The nurses in both regions were recruited through the distribution of flyers via email and the publication of the flyer in the association’s journal for nurses. The inclusion criteria for study participation were as follows: being at least 18 years old, holding a nursing degree, working in a Hong Kong or Swiss hospital during the survey period, having access to an electronic device and understanding either Chinese or German. Prior to participation, study information and consent for participation were obtained from all participants. The details of the study methodology have been described in our previous publication [50]. The previous study examined, with a structural equation model, whether psychological flexibility can mitigate the negative impact of low job satisfaction and high burnout levels on anxiety, depression, stress and well-being.

The present study extends the research question to examine whether psychological flexibility also serves as a buffer function for additional mental health outcomes (burnout), for physical health outcomes (insomnia) and for behavioral outcomes (turnover intention). Moreover, this study aims to investigate whether the mental health of nursing professionals differ across cultures.

### 2.2. Measures

Established psychological scales were used to assess constructs. Measures that did not exist in the local language (i.e., German or Chinese) were translated using the forward–backward translation by a native speaker in the respective target language. Well-validated measures of predictors, outcomes and questions regarding the impact of the pandemic on personal stress levels, absenteeism and turnover intentions were included in the survey after a consensus agreement among the members of this study. The questions regarding the impact of the pandemic on personal stress levels, absenteeism and turnover intentions were as follows: “Involved in COVID-19 treatment?”; “Feeling overly stressed/exhausted by work?”; “Increased feeling of stress since the outbreak of the pandemic?”; “Been absent from work or called sick sue to stress/exhaustion?”; “Been more absent from work or called sick since the outbreak of the pandemic?”; “Thinking of quitting the nursing profession since the outbreak of the pandemic?”.

### 2.3. Predictors

#### 2.3.1. Stress

Perceived Stress Scale (PSS-10). Stress was measured using the Perceived Stress Scale (PSS-10; [51]). The PSS assesses an individual’s appraisal of how stressful situations in their life are. Items ask about people’s feelings and thoughts during the last month. A total score is produced, with higher scores indicating greater overall distress. Cronbach’s alpha demonstrated good reliability of 0.84 [52]. In Hong Kong, Cronbach’s alpha for this scale was 0.72, while in Switzerland, it was 0.84.

#### 2.3.2. Job Satisfaction

McCloskey/Mueller Satisfaction Scale (MMSS) was used to measure job satisfaction. The inventory consists of 31 Likert scale questions with subscales for satisfaction with extrinsic reward, scheduling, family/work balance, co-workers, interactions, professional opportunities, praise/recognition and control/responsibility. The participants indicated their level of agreement with statements on a 5-point scale ranging from “very satisfied” to “very dissatisfied”. An example item is “How satisfied are you with recognition for your work from superiors?”. For each subscale, sum scores were calculated, with higher values indicating higher job satisfaction [53,54,55,56]. Cronbach’s alpha values for the overall MMSS scale in Hong Kong and Switzerland were 0.83 and 0.91, respectively. For the subscales, Cronbach’s alpha values varied between 0.77 and 0.93.

#### 2.3.3. Psychological Flexibility

The Psy-Flex questionnaire was used to measure the six processes of psychological flexibility (defusion, acceptance, present moment awareness, self as context, values, and committed action) [57]. The questionnaire consists of 6 Likert scale questions. The participants indicated their level of agreement with statements on a 5-point scale ranging from “very often” to “very seldom”. An example item is “even if I am somewhere else with my thoughts, I can focus on what’s going on in important moments”. A reversed sum score was used, with a higher score indicating greater psychological flexibility. In Hong Kong, Cronbach’s alpha for this scale was 0.83, while in Switzerland, it was 0.82.

### 2.4. Outcome Variables

#### 2.4.1. Burnout

Maslach Burnout Inventory (MBI-HSS-22, 7-point Likert scale) was used to measure three dimensions of burnout [58]. The inventory consists of 22 Likert scale questions with subscales of emotional exhaustion (EE), depersonalization (DP), and personal accomplishment (PA). The participants indicated their level of agreement with statements on a 7-point scale ranging from “never” to “daily”. An example item is “I feel drained at the end of a workday”. For each subscale, sum scores are calculated. High scores in EE and DP, as well as a lower score in PA, indicate high degrees of burnout [58]. Cronbach’s alphas of the EE, DP, and PA subscales in the Hong Kong and Switzerland sample were 0.79, 0.78, and 0.84, and 0.81, 0.73, and 0.81, respectively.

#### 2.4.2. Insomnia

Insomnia Severity Index (ISI; [59]) was used to measure the severity of insomnia. The screening tool consists of 7 Likert scale questions. The participants indicated their level of agreement with statements on a 5-point scale ranging from “none” to “very” or “not at all noticeable” to “very much noticeable”. An example item is “How noticeable to others do you think your sleeping problems is in terms of impairing the quality of your life?”.

Sum scores were calculated, and higher scores indicated severe sleep problems. Cronbach’s alphas in Hong Kong and Switzerland were 0.87 and 0.83, respectively.

#### 2.4.3. Turnover Intention

To assess turnover intention, participants were asked how often they were thinking of quitting the nursing profession since the outbreak of the pandemic. The participants indicated their level of turnover intention on a 5-point scale ranging from “never” to “all the time”.

### 2.5. Statistical Analysis

To examine whether there are differences in burnout, insomnia and turnover intention between nurses from an Eastern and a Western cultural context, the Chi-square test (χ^2^) and *t*-tests were used. To address the issue of different sample sizes between Hong Kong (n = 158) and Switzerland (n = 294), the homogeneity of variances was tested with the Levens’s test, and for the Chi-square test, it was checked whether the expected frequencies were sufficiently large.

To determine the predictive potential for psychological flexibility, five two-step hierarchical multiple regression analyses with the dependent variables emotional exhaustion, depersonalization, personal accomplishment, insomnia, and turnover intention were conducted. For all five regressions, age, gender, stress and job satisfaction were included as independent variables in step 1. In step 2, psychological flexibility was added as an independent variable. We used an alpha level of 0.05 for all statistical tests. Psychological flexibility was considered predictive if it explained variance over and above the variables in step 1. To assess whether the assumptions for regression were met, histograms and scatterplots were examined to evaluate linearity, the normal distribution of residuals and homoscedasticity. Additionally, the correlation matrix was inspected, revealing no signs of multicollinearity. The Durbin–Watson test was conducted to assess the independence of errors, and boxplots were used to identify outliners.

Statistical analyses were performed using IMB Statistics Version 27 (SPSS, Inc., Chicago, IL, USA).

## 3. Results

The demographic characteristics and the responses to the questions regarding the impact of the pandemic on personal stress levels, absenteeism and turnover intentions of the online survey are presented in Table 1. The mean age of the participants was 36.9 (SD = 10.9) years; participants worked in different hospital units, with many working on units with COVID-19 patients (62.4%). The average employment level was 80% (mean [SD] 79.23 [20.6]), and 85.2% were working shifts while participating in this study. A total of 62.4% of the respondents were involved in the treatment of COVID-19 patients, and 67% reported increased stress levels since the outbreak of the pandemic. A total of 27.2% of the participants stated that they were absent from work due to stress, with 15.3% of them indicating that they were absent more frequently due to stress since the pandemic began compared to before the outbreak. More than half of the respondents (52.5%) have thought about leaving the nursing profession since the outbreak of the pandemic sometimes (25.9%), fairly often (20.8%) or very often (5.8%). 

### 3.1. Comparison of Burnout, Insomnia and Turnover Intention Between Nurses from an Eastern and a Western Cultural Context

#### 3.1.1. Burnout

Nurses from Switzerland and Hong Kong both reported high levels of emotional exhaustion (51.8%) and depersonalization (40.5%) and low levels of personal accomplishment (61.7%). No difference between the two cultures was found for emotional exhaustion (t = 0.67, *p* = 0.506); however, nurses from Hong Kong had higher levels of depersonalization (60.8%) than nurses from Switzerland (29.6%) (t = −6.43, *p* = < 0.001) and lower levels of personal accomplishment (81.6%) than nurses from Switzerland (51%) (t = 8.9, *p* = < 0.001). A summary of the levels of burnout of the participating nurses is presented in Table 2.

#### 3.1.2. Insomnia

Almost half of all participants suffered from subthreshold insomnia, and 20% of the participants suffered from clinical insomnia with a moderate or severe level.

The mean ISI was 10.20 (5.25), conserved as subthreshold insomnia, by a possible maximal score of 28 within the scale. No region-specific differences were found. A third (n = 143) had no clinically significant insomnia, almost half had subthreshold insomnia (47.8%, n = 216), 18.4% (n = 83) had clinical insomnia considered as moderate severity and 2.2% (n = 10) had severe clinical insomnia.

#### 3.1.3. Absenteeism and Turnover Intention

Due to feelings of stress and exhaustion, 27.2% of the participants had called in sick or were absent from work in the past. Among these, 15.3% were absent from work more often since the beginning of the pandemic. Regarding increased absenteeism at work since the outbreak of the pandemic, nurses from Hong Kong have called in sick more often than nurses from Switzerland due to stress or exhaustion (25.9% versus 9.5%, *p* < 0.001).

Nurses from Switzerland reported higher turnover intentions than nurses from Hong Kong. In Switzerland more than 60% sometimes, fairly often or very often considered leaving the job, whereas in Hong Kong, 37% considered leaving (27.9% versus 22.2% for sometimes, 24.5% versus 13.9% for fairly often, 8.2% versus 1.3.% for very often, *p* < 0.001%). The absenteeism rate and turnover intentions during the pandemic of the participating nurses are presented in Table 1.

### 3.2. Predictive Potential of Psychological Flexibility

#### 3.2.1. Burnout

##### Emotional Exhaustion

The results of the regression for the dependent variable emotional exhaustion presented in Table 3 showed that the first model with the independent variables age, gender, stress and job satisfaction was significant for nurses from Switzerland ((F(4, 289) = 77.20), *p* < 0.001). A total of 52% of the variance for emotional exhaustion was explained by age, gender, job satisfaction and stress.

The second model, which included psychological flexibility, showed significant improvement from the first model for emotional exhaustion ((F_change_ (1, 288) = 6.75), *p* = 0.010) and explained 1% above the variance for emotional exhaustion than the first model.

In Hong Kong, the results of the regression for emotional exhaustion showed that the first model with the independent variables age, gender, stress and job satisfaction was significant for emotional exhaustion ((F (4, 153) = 10.25), *p* < 0.001). A total of 21% of the variance for emotional exhaustion was explained by age, gender, job satisfaction and stress. The second model, which included psychological flexibility, showed no significant improvement from the first model for emotional exhaustion.

##### Depersonalization

The results of the regression for the dependent variable depersonalization, presented in Table 4, showed that the first model with the independent variables age, gender, stress and job satisfaction was significant for nurses from Switzerland ((F(4, 289) = 24.68) *p* < 0.001). A total of 26% of the variance for depersonalization was explained by age, gender, job satisfaction and stress.

The second model, which included psychological flexibility, showed no significant improvement from the first model for depersonalization.

In Hong Kong, the results of the regression for depersonalization showed that the first model with the independent variables age, gender, stress and job satisfaction was not significant for depersonalization. The second model, which included psychological flexibility, was also not significant for depersonalization.

##### Personal Accomplishment

The results of the regression for the dependent variable personal accomplishment, presented in Table 5, showed that the first model with the independent variables age, gender, stress and job satisfaction was significant for nurses from Switzerland ((F(4, 289) = 9.06), *p* < 0.001). A total of 11% of the variance for personal accomplishment was explained by age, gender, job satisfaction and stress.

The second model, which included psychological flexibility, showed significant improvement from the first model for personal accomplishment ((F_change_(1, 288) = 7.06), *p* = 0.009) and explained 1% above the variance for personal accomplishment than the first model.

In Hong Kong, the results of the regression for personal accomplishment showed that the first model with the independent variables age, gender, stress and job satisfaction was significant for personal accomplishment ((F(4, 153) = 4.10), *p* = 0.003). A total of 10% of the variance for personal accomplishment was explained by age, gender, job satisfaction and stress. The second model, which included psychological flexibility, showed significant improvement from the first model for personal accomplishment ((F_change_ (1, 152) = 61.55), *p* < 0.001) and explained 26% more of the variance for personal accomplishment than the first model.

#### 3.2.2. Insomnia

The results of the regression for the dependent variable insomnia, presented in Table 6, showed that the first model with the independent variables age, gender, stress, and job satisfaction was significant for nurses from Switzerland ((F(4, 289) = 30.30), *p* < 0.001). A total of 30% of the variance for insomnia was explained by age, gender, job satisfaction and stress.

The second model, which included psychological flexibility, showed significant improvement from the first model for ((F_change_(1, 288) = 8.88), *p* = 0.003) and explained 2% above the variance for insomnia than the first model.

In Hong Kong, the results of the regression for insomnia showed that the first model with the independent variables age, gender, stress and job satisfaction was significant for insomnia ((F(4, 153) = 7.40), *p* < 0.001). A total of 16% of the variance for insomnia was explained by age, gender, job satisfaction and stress. The second model, which included psychological flexibility, showed no significant improvement from the first model for insomnia.

#### 3.2.3. Turnover Intention

The results of the regression for the dependent variable turnover intention, presented in Table 7, showed that the first model with the independent variables age, gender, stress and job satisfaction was significant for nurses from Switzerland ((F(4, 289) = 25.52), *p* < 0.001). A total of 26% of the variance for turnover intention was explained by age, gender, job satisfaction and stress.

The second model, which included psychological flexibility, showed no significant improvement from the first model for turnover intention.

In Hong Kong, the results of the regression for variable turnover intention, showed that the first model with the independent variables age, gender, stress and job satisfaction was significant for turnover intention ((F(4, 153) = 2.60) *p =* 0.038). A total of 6% of the variance for turnover intention was explained by age, gender, job satisfaction and stress. The second model, which included psychological flexibility, showed no significant improvement from the first model for turnover intention.

## 4. Discussion

The main aims of this study were to investigate cross-cultural differences in the prevalence of burnout, insomnia and turnover intention among nurses during the second wave of the COVID-19 pandemic and to examine the role of psychological flexibility as a protective factor against burnout, insomnia, and turnover intention.

Our results show that nurses from both regions suffer, and high levels of emotional exhaustion are present regardless of the region. At the time of the survey, the COVID-19 pandemic had resulted in approximately 52,000 infected individuals in Switzerland and 920 registered deaths as a result of the COVID-19 infection. In Hong Kong, during the same period, approximately 1190 COVID-10 infected individuals were registered, and 20 deaths were reported as a result of the COVID-19 disease [60].

However, Hong Kong has overall higher burnout rates than nurses form Switzerland, while turnover rates among nurses from Switzerland are higher than those of nurses from Hong Kong. Guo [27] and Fish [26] found that Australian nurses exhibited higher burnout rates compared to Chinese nurses, and Guo [27] reported no differences in turnover intentions between the two groups. These results stand in contrast to our findings, which indicate that nurses from the Eastern cultural background had overall higher burnout rates compared to their Western counterparts. The contrasting results may be explained by the fact that in the studies by Guo [27] and Fish [26], Eastern and Western cultures were compared between a developed and a developing country, whereas our study compared Eastern and Western cultures from both developed regions with comparable healthcare systems. In line with our findings, the study by Poghosyan [28], which compared burnout rates among nurses in six industrialized countries (U.S., Canada, U.K., Germany, New Zealand, and Japan), reported the highest burnout rates for nurses in Japan and the lowest for nurses in Germany. Variation in burnout levels in our study may partially be attributed to cultural differences between the two regions. In collectivist cultures, the well-being of the group, belonging to a group and group harmony are central values [24]. This can lead to nurses from Hong Kong putting their personal well-being aside and suppressing their emotions to meet social expectations and maintain harmony within the group. This, in turn, can result in high levels of internal stress and pressure, eventually leading to emotional detachment, which could explain the high depersonalization levels observed among nurses from Hong Kong in our study.

Additionally, it is known that collectivist cultures often have high role expectations [24]. These can create significant pressure when individuals must meet multiple expectations, such as the role of a family member, employee or member of society. Feelings of failure may arise when one is unable to meet these demands. This could explain why nurses from Hong Kong experience a low sense of personal accomplishment. Furthermore, in collectivist societies, group success is often prioritized over individual success [24]. As a result, nurses may perceive their personal contributions as less significant compared to those in individualistic cultures, where personal achievement is highly valued.

Moreover, there is often a large power distance between superiors and employees in collectivist cultures [24]. This can lead to feelings of overwhelm and a lack of autonomy, which can further contribute to depersonalization and a low sense of accomplishment. The fear of losing face or being judged negatively by colleagues or superiors may also cause nurses from Hong Kong to push themselves beyond their limits. This can lead to severe exhaustion, depersonalization and a low sense of personal accomplishment when they are unable to meet these high expectations.

In addition to the differences in burnout rates, differences in turnover intention were also observed between Hong Kong and Switzerland. Nurses in Switzerland reported higher turnover intentions than nurses from Hong Kong. This may also partly be explained by cultural differences between the two regions. For example, the degree of uncertainty avoidance can be mentioned as a cultural factor. It is known that in collectivist cultures, a high degree of uncertainty avoidance exists [24]. This could explain why nurses from Hong Kong perceive quitting their job, despite high levels of stress, as more significant and consider it less as an option compared to nurses from Switzerland. In Switzerland, as an individualistic culture, there is presumably a lower degree of uncertainty avoidance, making resignation a more conceivable option for nurses there than for those from Hong Kong. Additionally, prevailing role expectations and the fear of losing face may also contribute to the fact that nurses from Hong Kong are less likely to consider leaving their job, despite experiencing high levels of stress, compared to their Swiss counterparts. Another reason for the lower turnover intention among nurses in Hong Kong could be the high-performance pressure and fast-paced environment, where changing jobs is not necessarily seen as a way to reduce stress.

Regarding our second research aim, our results showed that psychological flexibility was predictive above and beyond the variance in Step 1, which included stress and job satisfaction as predictors for emotional exhaustion and personal accomplishment and also for insomnia in Switzerland. However, in Hong Kong, it was only predictive above and beyond the variance for Step 1 for personal accomplishment. The protective function of psychological flexibility against burnout can possibly be explained by the fact that individuals with higher levels of psychological flexibility are better able to tolerate difficult emotions in challenging situations, rather than avoiding them, which can lead to long-term suffering [61]. Moreover, they are more effective at distancing themselves from unhelpful thoughts, rather than fusing with them, which enables them to experience a greater sense of control. Individuals with higher levels of psychological flexibility are also more able to act according to their values, which may be another reason why nurses who are guided by their professional values are less likely to experience symptoms of burnout, as they derive satisfaction from the practice of their profession. This implies that psychological flexibility should be targeted in further public initiatives, as it is protective for nurses with different cultural backgrounds and different workforce contexts. These findings, which underpin the protective function of psychological flexibility across various contexts, are in line with previous research showing that psychological flexibility (PF) was a protective factor for the general working population across different workforce contexts [48,62,63]. Acceptance and commitment therapy (ACT) consists of six processes that are conceptualized as positive psychological skills and aim to promote PF [61]. The six core processes of ACT, including acceptance, cognitive defusion, being present, self as context, values and committed action, could be delivered as part of health programs through ACT training sessions or workshops conducted by ACT trainers. Previous studies have shown that ACT-based training programs are effective in managing work-related stress and may therefore provide a valuable framework for mental health promotion within healthcare organizations [64,65,66].

Additionally, it is important to note that, despite the differences we found between the two regions regarding burnout levels and turnover intention, nurses in both regions exhibit high levels of burnout, insomnia and turnover intention. This suggests a generally high level of stress among nursing staff and is in line with studies that were conducted during to the pandemic, indicating that a pandemic is a risk factor with significant implications for the mental health of nurses [67,68,69]. The burnout rates in our study are higher (51.8% of high emotional exhaustion, 40.5% of high depersonalization and 61.7% of low personal accomplishment) than those reported in a review of 16 surveys involving 18,935 nurses (34.6% for emotional exhaustion, 12.6.% for depersonalization and 15.2% for low personal accomplishment) [38] that included studies conducted during the COVID-19 pandemic prior to this study, suggesting that with increasing length of the pandemic, the burden and impact on mental health increased.

### Limitations

This study had several limitations. First, the cross-sectional design makes no causation between the variables possible. Longitudinal and experimental designs are required to verify these results. Second, as all data were obtained using self-reported measures, answers might be biased through social desirability, acquiescence or a tendency to extreme values.

Furthermore, beyond cultural factors, other variables, such as systemic factors like the presence of health policies and work-related factors like organizational support, may also contribute to explaining the differences in burnout rates and turnover intentions. However, these fall beyond the scope of the current study but should be explored in future research.

These limitations notwithstanding, this study demonstrated that a large group of nurses from two different cultural backgrounds reported moderate to high levels of burnout one year after the onset of the pandemic. These negative effects are not only relevant for individuals and their mental health but also for organizations and ultimately for society. The high turnover intention observed among nurses in both regions indicates that the risk of exacerbating the existing shortage of skilled nursing professionals is increasing. These findings have implications for public health initiatives. As nurses are particularly vulnerable to poor mental health, interventions aiming at improving mental health should be prioritized in their development. Promoting their mental health is relevant for society as a whole; otherwise, many nurses would no longer be able to maintain their profession. In times like the pandemic, this would dramatically impact public healthcare and worsen the already existing shortage of nurses. Improving the mental health of nurses should therefore be a high public health priority. Health interventions for nurses across regions should apply psychological processes, such as psychological flexibility. Our results identified psychological flexibility as a key factor that reduces the risk of developing mental health problems in nurses from Eastern and Western regions. Our results showed that cross-cultural differences in mental health and mental health associated issues among nurses exist, which suggest that tailored culturally relevant interventions may be more appropriate than universal approaches, as they can specifically address the respective issues with the highest priority.

## 5. Conclusions

This study found that burnout rates and turnover intention of nurses from two geographical regions are alarming and worse than previously assumed. Furthermore, cross-cultural differences were identified in this study, indicating higher overall burnout rates among nurses in Hong Kong and higher turnover intention rates among nurses in Switzerland. Given the limited available research on the impact of culture on the mental health of nursing professionals and the contrasting results of previous studies, our findings should be replicated among nurses from individualistic and collectivist cultural backgrounds to establish more definitive conclusions. Lastly, the study was able to show that psychological flexibility is a protective factor for burnout and turnover intentions across cultures. Public health initiatives that aim to improve nurses’ mental and physical health are needed. Psychological flexibility could be a key factor for such health initiatives and should therefore be further investigated concerning its effects on mental health in future studies.

## Figures and Tables

**Table 1 nursrep-15-00052-t001:** Sociodemographics of Swiss and Hong Kong nurses.

	Overall (N = 452)		Switzerland (n = 294)		Hong Kong (n = 158)		χ^2^ (df)/t	*p*
	N	%	n	%	n	%		
Age (years) (M, SD)	36.9 (10.9)		39.1 (11.5)		32.8 (8.5)		6.60	<0.001
Gender								
Male	70	15.5	36	12.2	34	21.5	6.75 (1)	0.009
Female	382	84.5	258	87.8	124	78.5		
Current working area								
Accident and Emergency	38	8.4	26	8.8	12	7.6	74.33 (13)	<0.001
Pediatric	19	4.2	4	1.4	15	9.5		
Obstetrics and Gynecology	13	2.9	5	1.7	8	5.1		
Internal Medicine	124	27.4	82	27.9	42	26.6		
Surgery	34	7.5	16	5.4	18	11.4		
Operating Theatre	18	4	10	3.4	8	5.1		
Rehabilitation	9	2	4	1.4	5	3.2		
Intensive Care Unit	28	6.2	24	8.2	4	2.5		
Out-Patient Clinics (general or special)	28	6.2	13	4.4	15	9.5		
Geriatrics	91	20.1	80	27.2	11	7.0		
Palliative	6	2	8	2.7	1	0.6		
Psychiatry	6	1.3	2	0.7	4	2.5		
Orthopedic	6	1.3	0	0	6	3.8		
Others	20	4.4	11	3.7	9	5.7		
Working experience in nursing								
<1 year	22	4.9	2	0.7	20	12.7	94.12 (4)	<0.001
1–3 years	38	8.4	13	4.4	25	15.8		
3–5 years	52	11.5	23	7.8	29	18.4		
5–10 years	98	21.7	55	18.7	43	27.2		
>10	242	53.5	201	68.4	41	25.9		
Work pensum (%), M (SD)	79.23 (20.6)		77.24 (21.7)		82.9 (17.9)		−2.82	0.005
Shift working								
Yes	385	85.2	259	88.1	126	79.7	5.67 (1)	0.017
No	67	14.8	35	11.9	32	20.3		
Involved in COVID-19 patient treatment								
Yes	282	62.4	217	73.8	65	41.1	46.75 (1)	<0.001
No	170	37.6	77	26.2	93	58.9		
Feeling overly stressed/exhausted by work								
Yes	339	75	226	79.9	113	71.5	1.57 (1)	0.210
No	113	25	68	23.1	45	28.5		
Increased feeling of stress since the outbreak of the pandemic?								
Yes	303	67	203	69	100	63.3	0.14 (1)	0.708
No	36	8	23	7.8	13	8.2		
Been absent from work or called sick due to stress/exhaustion?								
Yes	123	27.2	75	25.5	48	30.4	1.23 (1)	0.267
No	329	72.8	219	74.5	110	69.6		
Been more absent from work or called sick since the outbreak of the pandemic?								
Yes	69	15.3	28	9.5	41	25.9	27.48 (1)	<0.001
No	54	11.9	47	16	7	4.4		
Thinking of quitting the nursing profession since the outbreak of the pandemic?								
Never	127	28.1	60	20.4	67	42.4	33.10 (4)	<0.001
Almost never	88	19.5	56	19	32	20.3		
Sometimes	117	25.9	82	27.9	35	22.2		
Fairly often	94	20.8	72	24.5	22	13.9		
Very often	26	5.8	24	8.2	2	1.3		

Note. COVID-19 = coronavirus, M = mean; N = total sample number; n = number of samples per group; SD = standard deviation; χ2 = Chi-square; t = t-statistics.

**Table 2 nursrep-15-00052-t002:** Burnout, insomnia and psychological flexibility of Swiss and Hong Kong nurses.

	Overall (N = 452)		Switzerland (n = 294)		Hong Kong (n = 158)		χ^2^ (df)/t	*p*
	N	%	n	%	n	%		
Burnout								
Emotional exhaustion, M (SD)	25.78 (10.23)		25.99 (11.08)		25.37 (8.44)		0.67	0.506
Low	109	24.1	79	26.9	30	19	9.80 (2)	0.007
Moderate	109	24.1	58	19.7	51	32.3		
High	234	51.8	157	53.4	77	48.7		
Depersonalization, M (SD)	8.50 (5.74)		7.27 (5.46)		10.78 (5.56)		−6.43	<0.001
Low	158	35	127	43.2	31	19.6	43.41 (2)	<0.001
Moderate	111	24.6	80	27.2	31	19.6		
High	183	40.5	87	29.6	96	60.8		
Personal accomplishment, M (SD)	30.36 (7.53)		32.62 (6.30)		26.17 (7.85)		8.9	<0.001
Low	279	61.7	150	51	129	81.6	40.84 (2)	<0.001
Moderate	134	29.6	111	37.8	23	14.6		
High	39	8.6	33	11.2	6	3.8		
Insomnia								
ISI total score, M (SD)	10.20 (5.26)		10.62 (5.40)		9.44 (4.50)		2.30	0.023
No clinically significant insomnia	143	31.6	85	28.9	52	32.9	5.10 (3)	0.17
Subthreshold insomnia	216	47.8	139	47.3	78	49.4		
Clinical insomnia—moderate severity	83	18.4	63	21.4	28	17.7		
Clinical insomnia—severe	10	2.2	7	2.4	0	0		
Psychological flexibility, M (SD)	21.38 (4.76)		23.02 (4.29)		18.33 (4.06)		11.3	<0.001

Note. COVID-19 = coronavirus, M = mean; N = total sample number; n = number of samples per group; SD = standard deviation; χ^2^ = Chi-square; t = t-statistics.

**Table 3 nursrep-15-00052-t003:** Results of the two-step hierarchical regression analysis predicting emotional exhaustion.

	Switzerland (n = 294)				Hong Kong (n = 158)			
	β	SE	*p*	R^2^/ΔR^2^	β	SE	*p*	R^2^/ΔR^2^
Step 1 ^a^								
Age	−0.14	0.04	<0.001		0.03	0.72	0.11	
Gender ^b^	−0.40	1.40	0.78		0.18	1.49	0.90	
Stress	0.65	0.08	<0.001		0.81	0.14	<0.001	
Job satisfaction	−0.26	0.03	<0.001	0.52	−0.01	0.05	0.77	0.21
Step 2 ^c^								
Age	−0.13	0.04	0.001		0.02	0.07	0.74	
Gender ^b^	−0.23	1.40	0.83		0.01	1.49	1.00	
Stress	0.53	0.10	<0.001		0.82	0.14	<0.001	
Job satisfaction	−0.26	0.03	<0.001		−0.03	0.05	0.90	
Psy-Flex	−0.33	0.13	0.010	0.53/0.01	0.26	0.15	0.09	0.23/0.02

Note. ^a^ Step 1 includes Predictor 1 (stress PSS), Predictor 2 (job satisfaction MMSS), covariates included age, gender (male/female), ^b^ dummy coded (male = 0, female = 1), ^c^ Step 2 includes Step 1 and the added predictor psychological flexibility (Psy-Flex).

**Table 4 nursrep-15-00052-t004:** Results of the two-step hierarchical regression analysis predicting depersonalization.

	Switzerland (n = 294)				Hong Kong (n = 158)			
	β	SE	*p*	R^2^/ΔR^2^	β	SE	*p*	R^2^/ΔR^2^
Step 1 ^a^								
Age	−0.14	0.02	<0.001		0.02	0.05	0.67	
Gender ^b^	−3.46	0.86	<0.001		−2.15	1.08	0.05	
Stress	0.08	0.05	0.10		0.10	0.10	0.33	
Job satisfaction	−0.08	0.02	<0.001	0.26	−0.05	0.03	0.15	0.05
Step 2 ^c^								
Age	−0.14	0.02			0.03	0.05	0.63	
Gender ^b^	−3.46	0.86	<0.001		−2.10	1.10	0.05	
Stress	0.09	0.06	0.12		0.10	0.10	0.40	
Job satisfaction	−0.08	0.02	<0.001		−0.04	0.03	0.20	
Psy-Flex	0.02	0.08	0.81	0.26/0.00	−0.09	0.11	0.41	0.05/0.00

*Note*. ^a^ Step 1 includes Predictor 1 (stress PSS), Predictor 2 (job satisfaction MMSS), covariates included age, gender (male/female), ^b^ dummy coded (male = 0, female = 1), ^c^ Step 2 includes Step 1 and the added predictor psychological flexibility (Psy-Flex).

**Table 5 nursrep-15-00052-t005:** Results of the two-step hierarchical regression analysis predicting personal accomplishment.

	Switzerland (n = 294)				Hong Kong (n = 158)			
	β	SE	*p*	R^2^/ΔR^2^	β	SE	*p*	R^2^/ΔR^2^
Step 1 ^a^								
Age	0.02	0.03	0.50		−0.03	0.07	0.64	
Gender ^b^	1.33	1.10	0.22		0.79	1.48	0.59	
Stress	−0.22	0.07	0.001		−0.04	0.14	0.78	
Job satisfaction	0.05	0.02	0.01	0.11	0.17	0.05	0.0000	0.10
Step 2 ^c^								
Age	0.02	0.03	0.59		−0.07	0.06	0.27	
Gender ^b^	1.26	1.07	0.24		0.09	1.26	0.94	
Stress	−0.12	0.07	0.10		0.02	0.12	0.87	
Job satisfaction	0.05	0.02	0.02		0.12	0.04	0.002	
Psy-Flex	0.26	0.10	0.009	0.13/0.02	1.01	0.13	<0.001	0.36/0.26

*Note.* ^a^ Step 1 includes Predictor 1 (stress PSS), Predictor 2 (job satisfaction MMSS), covariates included age, gender (male/female), ^b^ dummy coded (male = 0, female = 1), ^c^ Step 2 includes Step 1 and the added predictor psychological flexibility (Psy-Flex).

**Table 6 nursrep-15-00052-t006:** Results of the two-step hierarchical regression analysis predicting insomnia.

	Switzerland (n = 294)				Hong Kong (n = 158)			
	β	SE	*p*	R^2^/ΔR^2^	β	SE	*p*	R^2^/ΔR^2^
Step 1 ^a^								
Age	0.02	0.02	0.33		0.01	0.04	0.78	
Gender ^b^	−0.57	0.83	0.45		−0.72	0.90	0.46	
Stress	0.35	0.05	<0.001		0.36	0.08	<0.001	
Job satisfaction	−0.06	0.02	<0.001	0.30	−0.05	0.03	0.09	0.16
Step 2 ^c^								
Age	0.02	0.02	0.26		0.01	0.04	0.78	
Gender ^b^	−0.50	0.81	0.54		−0.66	0.90	0.46	
Stress	0.27	0.06	<0.001		0.36	0.08	<0.001	
Job satisfaction	−0.06	0.02	<0.001		−0.04	0.03	0.12	
Psy-Flex	−0.22	0.07	0.003	0.32/0.02	−0.08	0.09	0.37	0.17/0.00

*Note*. ^a^ Step 1 includes Predictor 1 (stress PSS), Predictor 2 (job satisfaction MMSS), covariates included age, gender (male/female), ^b^ dummy coded (male = 0, female = 1), ^c^ Step 2 includes Step 1 and the added predictor psychological flexibility (Psy-Flex).

**Table 7 nursrep-15-00052-t007:** Results of the two-step hierarchical regression analysis predicting turnover intention.

	Switzerland (n = 294)				Hong Kong (n = 158)			
	β	SE	*p*	R^2^/ΔR^2^	β	SE	*p*	R^2^/ΔR^2^
Step 1 ^a^								
Age	−0.01	0.01	0.25		−0.02	0.01	0.04	
Gender ^b^	−0.08	0.20	0.69		−0.01	0.22	0.95	
Stress	0.04	0.01	0.001		0.02	0.02	0.41	
Job satisfaction	−0.03	0.00	<0.001	0.26	−0.01	0.01	0.11	0.06
Step 2 ^c^								
Age	− 0.01	0.01	0.27		−0.02	0.01	0.04	
Gender ^b^	−0.08	0.20	0.70		0.01	0.22	0.98	
Stress	0.04	0.01	0.007		0.01	0.02	0.45	
Job satisfaction	−0.03	0.00	<0.001		−0.01	0.01	0.16	
Psy-Flex	−0.01	0.02	0.56	0.26/0.00	−0.03	0.02	0.20	0.07/0.01

Note. ^a^ Step 1 includes Predictor 1 (stress PSS), Predictor 2 (job satisfaction MMSS), covariates included age, gender (male/female), ^b^ dummy coded (male = 0, female = 1), ^c^ Step 2 includes Step 1 and the added predictor psychological flexibility (Psy-Flex).

## Data Availability

The data supporting this study’s findings are available from the corresponding author upon reasonable request.

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
