# Peer review of "Cross-Cultural Comparison of Burnout, Insomnia and Turnover Intention Among Nurses in Eastern and Western Cultures During the COVID-19 Pandemic: Protective and Risk Factors"

_nursrep, 2025, doi:10.3390/nursrep15020052_

Round 1

Reviewer 1 Report

Comments and Suggestions for Authors

In detail, I read and analysed the manuscript Cross-cultural Comparison of Burnout, insomnia and Turnover Intention among Nurses in Eastern and Western Cultures: Protective and Risk Factors.

The manuscript deals with the well-researched but always relevant topics of burnout syndrome, insomnia, and turnover. The advantage of the study is that it included protective factors in the research.

Title

Given that the study was conducted during the COVID-19 pandemic, I suggest it be included in the title.

Abstract

An abstract is concise and well-written.

The introduction follows a funnel approach and ends by clearly formulating the study’s aims and questions. However, I suggest shortening the introduction. It is written too broadly and burdened with unnecessary statistical/epidemiological data. Also, I suggest that in line 97, the word little staff training be replaced with insufficient staff training, and then little mental support should be replaced with inadequate mental support.

Materials and Methods

Participants and procedure

For better readability, I recommend that the authors list the criteria for including and excluding nurses in the study, and please correct the typographical error in line 221.

Outcomes variables

Indicate whether the author’s consent was obtained for the questionnaire.

Results are presented in seven tables, are textual, and are interpreted appropriately and consistently throughout the manuscript. Please correct the typographical error in lines 377,420, 540-541 and 556. It is a disadvantage that you did not include the country of origin in the hierarchical analysis. In this way, you could confirm whether the country of origin significantly predicts the observed variable.

Discussion is thorough and provides an in-depth analysis of the results. However, I suggest that in line 599, the word goal be replaced with the aim, as well as in line 654, and a correction in writing COVID-19 in line 601.

The limitations of the study are listed and well described.

Conclusions are drawn coherently and supported by the listed citations.

The cited references are relevant, up-to-date and properly cited.

Final Comments

Congratulations to the authors on their manuscript. I believe these recommendations will strengthen the manuscript and improve its overall quality.

Author Response

Comments 1: Title – Given that the study was conducted during the COVID-19 pandemic, I suggest it be included in the title.

Response 1: Thank you very much for your input. We have changed the title to: “Cross-cultural comparison of Burnout, insomnia and turnover intention among nurses in Eastern and Western cultures during the Covid-19 pandemic: Protective and risk factors” (page 1, lines 2-4 in revised manuscript).

Comments 2: Abstract – An abstract is concise and well-written.

Response 2: Thank you very much.

Comments 3: The introduction follows a funnel approach and ends by clearly formulating the study’s aims and questions. However, I suggest shortening the introduction. It is written too broadly and burdened with unnecessary statistical/epidemiological data. Also, I suggest that in line 97, the word little staff training be replaced with insufficient staff training, and then little mental support should be replaced with inadequate mental support.

Response 3: Thank you for your suggestion. We have shortened the introduction to 2.5 pages by shortening and rewriting the following sections. In red, you will find the newly added section of the manuscript's introduction.

Pages 1, 2, first paragraph 1,  lines 36-60 in revised manuscript: Healthcare professionals, particularly nurses, face significant mental and physical burdens in their profession, with burnout being one of the most frequent reported consequences [1-3]. Burnout, a psychological syndrome resulting from prolonged occupational stress, is characterized by three dimensions: emotional exhaustion, depersonalization, and reduced professional efficacy [4,5]. Emotional exhaustion manifest as chronic fatigue and stress, while depersonalization reflects a detached and impersonal approach to others. Reduced professional efficacy denotes a diminished sense of accomplishment in one`s work [3,5-7].

High burnout rates among nurses were found even before the COVID-19 pandemic, with a meta-analysis reporting a global prevalence of burnout symptoms at 11.2% [8]. During the pandemic, burnout rates increased greatly, particularly among emergency nurses, with nearly 58% reporting high emotional exhaustion compared to 37% pre-pandemic ([9,10]. Sleep disorders, another prevalent issue, affected 38% of healthcare workers during the pandemic, with frontline nurses showing the highest rates [11,12].

The mental and physical health challenges not only impact individual nurses but also have systemic consequences, including increased turnover intentions. Pre-pandemic studies revealed a pooled turnover rate of 18% among nurses [13], with surged post-pandemic, with some studies reporting turnover intentions as high as 78% [14-17]. This exacerbates the global nursing shortage, with the World Health Organization projecting a deficit of 7.6 million nurses by 2030 [18].

Therefore, it is important to know both risk and protective factors that contribute to improving mental and physical health of nurses to increase the nurses’ commitment to their job and to counteract the current tendency of resigning or leaving the nursing profession at an early stage.”

Page 2, paragraph 2, lines 63-70 in revised manuscript: “Context specific factors such as cultural, social or work-related aspects, influence the mental health of nursing professionals, contributing to regional variations in well-being. Poor working conditions – high workload, long hours, low pay, lack of resources, and insufficient legal regulation- negatively impact nurses ‘mental health. The pandemic exacerbated these conditions, with increased workloads, fear of infection, and insufficient support affecting healthcare workers. A cross-country study found that nurses in regions with inadequate protective equipment, poor staff training, and limited mental support reported worse mental health than those in better supported regions. [1,19-22].”

Pages 2-3, paragraph 2,  lines 91-95 / 223-224 in revised manuscript: “Understanding these context-specific factors is crucial for tailoring health programs to regional needs. Despite its importance, cross-cultural studies on nurses ‘mental health remain limited, with existing research primarily comparing Western countries. Only a few studies have directly compared burnout rates between Western and Eastern cultures. The results of the studies indicate that differences between cultures exists; however, some studies found higher burnout values for Western countries, while other found higher values for Eastern countries [25-27].”

Comments 4: Materials and MethodsParticipants and procedure

For better readability, I recommend that the authors list the criteria for including and excluding nurses in the study, and please correct the typographical error in line 221.

Response 4: Many thanks for your input. For better readability, we have added the following addition (page 4, lines 336-240 in revised manuscript): “The inclusion criteria for study participating were as follows: being at least 18 years old, holding a nursing degree, working in a Hong Kong or Swiss hospital during the survey period, having access to an electronic device and understanding either Chinese or German. Prior to participation, study information and consent for participation was obtained from all participants.” In addition, the typo of the word inclusion in line 221 has been corrected.

Comments 5: Outcomes variables – Indicate whether the author’s consent was obtained for the questionnaire.

Response 5: Thank you for your reminder. With the purchase of the questionnaires we obtained the right to use them for research purposed.

Comments 6: Results are presented in seven tables, are textual, and are interpreted appropriately and consistently throughout the manuscript. Please correct the typographical error in lines 377,420, 540-541 and 556. It is a disadvantage that you did not include the country of origin in the hierarchical analysis. In this way, you could confirm whether the country of origin significantly predicts the observed variable.

Response 6: Thank you for indicating the typographical errors in lines 377, 420, 540-541 and 556, all of which are corrected in the revised manuscript.

We included potential risk and protective factors as predictors in the analysis. By running the regression for the entire sample as well as separately for Hong Kong and Switzerland, we were able to identify whether the predictors (job satisfaction, stress, age, gender, and psychological flexibility) had different effects on nurses in Hong Kong and Switzerland why we assumed that the inclusion of the country of origin as an additional predictor was obsolete.

Comments 7: Discussion is thorough and provides an in-depth analysis of the results. However, I suggest that in line 599, the word goal be replaced with the aim, as well as in line 654, and a correction in writing COVID-19 in line 601.

Response 7: Thank you for your suggestions for alternative wording and writing. We have made the changes according to your suggestions as follows.

On page 19, lines, 781, 783 in revised manuscript: “The main aims of this study were to investigate cross-cultural differences in the prevalence of burnout, insomnia and turnover intentions among nurses during the second wave of the COVID-19 pandemic and to examine the role of psychological flexibility as a protective factor against burnout, insomnia, and turnover intention.”

On page 20, line 845 in revised manuscript: “Regarding our second research aim, our results showed that psychological flexibility was predictive above and beyond the variance in Model one, which included stress and job satisfaction as predictors for emotional exhaustion and personal accomplishment and also for insomnia in Switzerland.”

Comments 8: The limitations of the study are listed and well described. 

Response 8: Thank you very much.

Comments 9: Conclusions are drawn coherently and supported by the listed citations.

Response 9: Thank you very much.

Comments 10: The cited references are relevant, up-to-date and properly cited.

Comments 10: Thank you very much.

Reviewer 2 Report

Comments and Suggestions for Authors

Dear Author,

It is delightful to have the opportunity to review " Cross-cultural comparison of Burnout, insomnia and turnover intention among nurses in Eastern and Western cultures: Pro-tective and risk factors"

Congratulations, the article is very well-written and structured.

I present suggestions and aspects that need to be clarified:

1.       Line 18 – Replace "turnout" with "turnover."

2.       Lines 38-39 – The definition of the burnout concept should be supported by additional references, such as: WHO. ICD-11. 11th Revision of the International Classification of Diseases. November 2020. Available online: https://icd.who.int/en

3.       Burnout Dimensions – While the manuscript cites several studies demonstrating that turnover is a pressing issue and provides statistics, for example: "They showed high rates of emotional exhaustion (26%), depersonalization (35%), and a lack of personal accomplishment (27%)" It is crucial to define the three dimensions of burnout explicitly: Emotional Exhaustion, Depersonalization, and Personal Accomplishment.

4.       Line 71 – The statement, "It was already known before the pandemic that nursing was one of the professions with the highest turnover intention rates," requires references to support this claim.

5.       Lines 218-221 – The description of COVID-19 statistics ("During this period there were approximately 52,000 COVID-19 infected individuals in Switzerland and 920 registered deaths. In Hong Kong, during the same period, approximately 1,190 COVID-19 infected individuals and 20 deaths were reported") seems out of place in the methodology section. Consider moving it to the introduction or results, where it could better contextualize the study.

6.       Table 1 (Sociodemographics) – This table presents results and should not be part of the methodology section.

7.       The substantial difference in sample sizes (n=294 for Switzerland vs. n=158 for Hong Kong) could impact the robustness of the t-test and chi-square test results. This methodological concern should be explicitly addressed in the limitations section.

8.       Was any method used to identify and manage outliers that could influence regression coefficients? This information should be included in the methodology.

9.       The manuscript must detail how the assumptions of hierarchical regression were assessed. For instance, clarify if normality, linearity, multicollinearity, and homoscedasticity were evaluated.

10.   Line 717 – Correct the typo "turno ver" to "turnover."

11.   The discussion inadequately addresses how health policies or working conditions, such as organizational support or workload, might influence burnout, insomnia, and turnover intention. These factors could be confounding variables impacting the study results, and their role should be explored.

12.   The discussion lacks specificity about strategies to promote psychological flexibility.

13.   While turnover intention is identified as a critical issue, the discussion does not fully explore the potential reasons for cultural differences beyond cultural explanations.

Best regards,

Author Response

Comments 1: Line 18 – Replace "turnout" with "turnover."

Response 1: Thank you for pointing this out. The typo has been corrected on page 1, line 18.

Turnover

Comments 2: Lines 38-39 – The definition of the burnout concept should be supported by additional references, such as: WHO. ICD-11. 11th Revision of the International Classification of Diseases. November 2020. Available online: https://icd.who.int/en

Response 2: Thank you for pointing this out. I agree with this comment and have included the ICD-11 as a reference for burnout on page 1, line 40.

Burnout, a psychological syndrome resulting from prolonged occupational stress, is characterized by three dimensions: emotional exhaustion, depersonalization, and reduced professional efficacy [4,5]. Emotional exhaustion manifests as chronic fatigue and stress, while depersonalization reflects a detached and impersonal approach to others. Reduced professional efficacy denotes a diminished sense of accomplishment in one`s work [3,5-7].

Comments 3: Burnout Dimensions – While the manuscript cites several studies demonstrating that turnover is a pressing issue and provides statistics, for example: "They showed high rates of emotional exhaustion (26%), depersonalization (35%), and a lack of personal accomplishment (27%)" It is crucial to define the three dimensions of burnout explicitly: Emotional Exhaustion, Depersonalization, and Personal Accomplishment.

Response 3: Agree, the three dimensions have been described explicitly on page 1,2, lines 40-44.

Emotional exhaustion manifests as chronic fatigue and stress, while depersonalization reflects a detached and impersonal approach to others. Reduced professional efficacy denotes a diminished sense of accomplishment in one`s work [3,5-7].

Comments 4: Line 71 – The statement, "It was already known before the pandemic that nursing was one of the professions with the highest turnover intention rates," requires references to support this claim.

Response 4: Agree, the sentence has been revised and added with new references on page 2, lines 52-54.

Pre-pandemic studies revealed a pooled turnover rate of 18% among nurses [13], with surged post-pandemic, with some studies reporting turnover intentions as high as 78% [14-17].  

Comments 5: Lines 218-221 – The description of COVID-19 statistics ("During this period there were approximately 52,000 COVID-19 infected individuals in Switzerland and 920 registered deaths. In Hong Kong, during the same period, approximately 1,190 COVID-19 infected individuals and 20 deaths were reported") seems out of place in the methodology section. Consider moving it to the introduction or results, where it could better contextualize the study.

Response 5: Thanks for pointing this out. I moved it to the discussion section as it fit best there, page 19, lines 787-791.

At the time of the survey the pandemic approximately 52`000 COVID-19 infected individuals in Switzerland and 920 registered deaths as a result of the COVID-19 infection. In Hong Kong, during the same period, approximately 1190 COVID-10 infected individuals were registered and 20 deaths were reported as a result of the COVID-19 disease [56].

Comments 6: Table 1 (Sociodemographics) – This table presents results and should not be part of the methodology section.

Response 6: Agree, the table has been relocated to the results section, page 7-10.

Comments 7: The substantial difference in sample sizes (n=294 for Switzerland vs. n=158 for Hong Kong) could impact the robustness of the t-test and chi-square test results. This methodological concern should be explicitly addressed in the limitations section.

Response 7: Thank you for pointing this out. To ensure that the results are not biased despite differing sample size, variance equality was tested for t-test, and it was confirmed that the expected frequencies were sufficiently large for the Chi-square test. I have added this explanation to the methods section, statistical analysis on page 6, lines 464-467 and therefore did not list it as a limitation.

To address the issue of different sample size between Hong Kong (n = 158) and Switzerland (n = 294), the homogeneity of variances was tested with the Levens`s test, and for the Chi-square test, it was checked whether the expected frequencies were sufficiently large.

Comments 8: Was any method used to identify and manage outliers that could influence regression coefficients? This information should be included in the methodology.

Response 8: Thanks for pointing this out. Yes, outliners were identified with boxplots. I have added this to the method section, statistical analysis on page 6, line 479.

The Durbin-Watson test was conducted to assess the independence of errors, and boxplots were used to identify outliners.

Comments 9: The manuscript must detail how the assumptions of hierarchical regression were assessed. For instance, clarify if normality, linearity, multicollinearity, and homoscedasticity were evaluated.

Response 9: Agree, I have added or the missing assumptions and how they were tested on page 6, statistical analysis, lines 475-479.

To access whether the assumptions for regression were met, histograms and scatterplots were examined to evaluate linearity, the normal distribution of residuals, and homoscedasticity. Additionally, the correlation matrix was inspected, revealing no signs of multicollinearity. The Durbin-Watson test was conducted to assess the independence of errors, and boxplots were used to identify outliners

Comments 10: Line 717 – Correct the typo "turno ver" to "turnover."

Response 10: Thank you for pointing this out. The typo has been corrected on page , Line 29, line 924.

turnover

Comments 11: The discussion inadequately addresses how health policies or working conditions, such as organizational support or workload, might influence burnout, insomnia, and turnover intention. These factors could be confounding variables impacting the study results, and their role should be explored.

Response 11: I agree. However, this point was addressed in the limitations section. However, I have now elaborated on it further in the limitation section and provided specific examples of additional influencing factors. Page 21, lines 895 – 899.

Furthermore, beyond cultural factors, other variables, such as systemic factors like the presence of health policies and work-related factors like organizational support, may also contribute to explaining the differences in burnout rates and turnover intentions. However, these fall beyond the scope of the current study but should be explored in future research.

Comments 12: The discussion lacks specificity about strategies to promote psychological flexibility. 

Response 12: Thanks for pointing this out. I have added to the discussion section how psychological flexibility can be improved. Page 21, lines 867-875.

The acceptance and commitment therapy (ACT) consists of six processes that are conceptualized as positive psychological skills and aim to promote psychological flexibility (PF) [57]. The six core processes of ACT, including acceptance, cognitive defusion, being present, self as context, values and committed action, and could be delivered as part of health programs through ACT training sessions or workshops conducted by ACT trainers. Previous studies have shown that ACT-based training programs are effective in managing work related stress and may therefore provide a valuable framework for mental health promotion within healthcare organizations [60-62].

Comments 13: While turnover intention is identified as a critical issue, the discussion does not fully explore the potential reasons for cultural differences beyond cultural explanations.

Response 13: Thanks for pointing this out. I have included an additional reason for potential differences between cultures regarding the varying levels of turnover intentions. Page 20, lines 843-845.

Another reason for the lower turnover intention among nurses in Hong Kong could be the high performance pressure and fast-paced environment, where changing jobs is not necessarily seen as a way to reduce stress.